# Electrochemical C−H deuteration of pyridine derivatives with D₂O

Zhiwei Zhao[1], Ranran Zhang[1], Yaowen Liu[1], Zile Zhu[1], Qiuyan Wang[1] ✉ & Youai Qiu ✉[1] ✉

Herein, we develop a straightforward, metal-free, and acid-/base-free electrochemical C4-selective C − H deuteration of pyridine derivatives with economic and convenient D₂O at room temperature. This strategy features an efficient and environmentally friendly approach with high chemo- and regioselectivity, affording a wide range of D-compounds, such as pyridines, quinolones, N-ligands and biorelevant compounds. Notably, the mechanistic experiments and cyclic voltammetry (CV) studies demonstrate that N-butyl-2-phenylpyridinium iodide is a crucial intermediate during the electrochemical transformation, which provides a general and efficient way for deuteration of pyridine derivatives.

Deuterium-labelled molecules are a critical kind of organic compounds, which have been widely applied into various research areas, such as elucidating the reaction mechanisms[1,2], isotopic tracer techniques[3–5], and pharmaceutical chemistry[6–10]. For example, N-heteroarenes, which are frequently employed as bioactive molecules and drugs for exploring the pharmacokinetic and pharmacodynamic (PK/PD) properties[11,12]. Therefore, it is unsurprising that the synthesis and application of deuterium-labeled (N-hetero)arenes has attracted significant attentions[13–18], especially for the D-labeled pyridine derivatives[19–21]. Among the various protocols that have been developed, the direct, simple and efficient H/D exchange strategy[22–24] stands prior to dehalogenative deuteration of halides and pseudohalides[25–27], owing to the readily available starting materials, their cost, as well as high atom economy, however, the challenge would be the selectivity control of the transformations. Despite the established C−H deuteration of pyridine derivatives in the presence of Brønsted/Lewis acid[28,29], base[30,31], or transition-metal[32,33] (Fig. 1A), the pursuance of methodologies with higher selectivity and D-incorporation under milder conditions with easy operation procedures is still on the way. Notably, McNally and co-workers developed a C4-selective C−H deuteration of pyridine derivatives through two-step process, which undergoing the heterocyclic phosphonium salts intermediator, and then forming the final deterateted pyridine derivatives with assistance of base, achieving high selective deuteration of pyridine derivatives (Fig. 1B)[34]. Furthermore, the development of approaches for site-selective C−H deuteration of pyridine derivatives

in a straightforward, sustainable, and efficient way is still highly desirable and challenging.

In recent years, the emergence of electrochemistry has brought opportunities and development in organic synthesis[35–52]. Electrosynthesis is becoming increasingly popular and considered as one of sustainable and desirable methodology that could substitute some traditional synthetic methods. We envisage that direct and selective C−H deuteration of pyridine derivatives can be achieved through the usage of electrochemistry. However, compared with well-developed electrooxidative C−H functionalization[53–65], electroreductively driven C−H functionalization of arenes have thus not been well elucidated[66–69]. Especially, selective C−H deuteration of pyridine derivatives under electroreductive conditions has not achieved yet.

In sharp contrast, with our continuous interests in sustainable electroreductively driven C−H functionalization[66], and electrochemical deuteration[70,71], herein, we would like to report our effort in developing a general, direct and efficient electrochemical C4-selective C−H deuteration of pyridine derivatives through reductive activation with economical D₂O, via a crucial N-butyl-2-phenylpyridinium iodide intermediate (Fig. 1C). This protocol offered a wide range of D-pyridine derivatives with high chemo- and regioselectivity in excellent yields. The salient features of this transformation including: (a) electroreductively driven C−H deuteration; (b) good to excellent D-incorporation; (c) metal-, acid-, or base-free process; (d) high chemo- and regioselectivity; (e) late-stage modification of N-ligands and biorelevant compounds.

[1]State Key Laboratory and Institute of Elemento-Organic Chemistry, Frontiers Science Center for New Organic Matter, College of Chemistry, Nankai University, Tianjin, China. ✉e-mail: qywang@nankai.edu.cn; qiuyouai@nankai.edu.cn

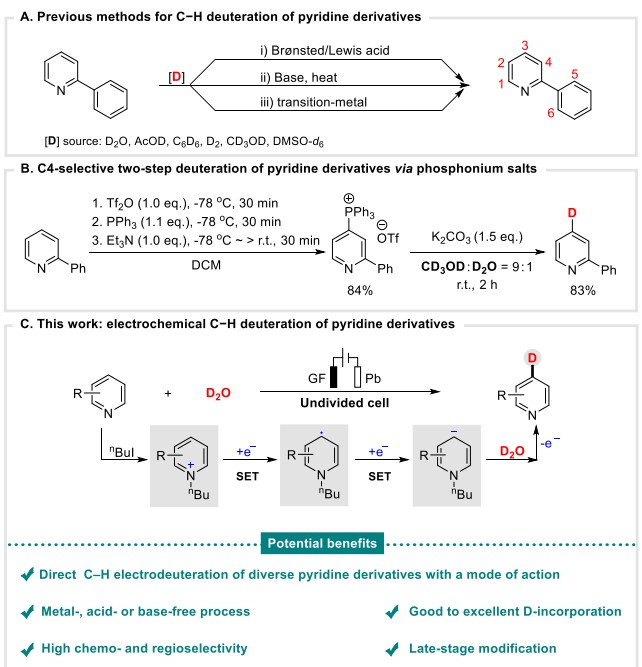

**Fig. 1 | Background and current work of deuteration for pyridine derivatives. A** Previous methods for C−H deuteration of pyridine derivatives. **B** C4-selective two-step deuteration of pyridine derivatives via phosphonium salts. **C** This work: electrochemical C−H deuteration of pyridine derivatives.

## Results

### Optimization of reaction conditions

Our investigations began by evaluating the C−H deuteration of pyridine substrates with deuterium oxide ($D_2O$). After some preliminary experiments, 2-phenylpyridine (**S1**) was selected as the model starting substrate for optimizing the reaction conditions. Encouragingly, only a single product **1** was observed, which proved that the reaction has high regioselectivity. After a careful selection of the system parameters, we obtained the optimal conditions for deuteration of **S1** with $D_2O$. Combining the reactants with electrolyte $^nBu_4NI$ in an undivided cell with DMF under constant current (20 mA) at room temperature for 10 hours, resulted in 99% yield and >99% deuterium incorporation (abbreviated as D-inc hereafter) (Table 1, entry 1). Initially, a variety of electrolytes were investigated. $^nBu_4NBF_4$ and $Et_4NI$ worked well and resulted in 99% and 90% D-inc of product **1** respectively (Table 1, entries 2 and 3), but no product was detected when $LiClO_4$ or NaI was employed (Table 1, entries 4 and 5). Then, an attempted to use NaOAc as a base, led to a decrease in deuteration (Table 1, entry 6). We also examined various solvents, such as DMA and MeCN, which afforded 82% and <5% D-inc of product **1** respectively (Table 1, entries 7 and 8). Next, we investigated various electrodes, including CF(+)|Pb(−), GF(+)|GF(−), and GF(+)|Pt(−) (Table 1, entries 9−11). However, the D-inc were <80%. Furthermore, we probed the effect of the current on this reaction. With a lower current, deuteration of **1** decreased significantly (Table 1, entries 12−13). The system also worked smoothly at higher temperatures (Table 1, entry 14, 50 °C). In addition, we found that this system was insensitive to the atmosphere and still gave excellent yield and D-inc under argon atmosphere (Table 1, entry 15). Finally, some control experiments proved that the presence of electricity and electrolyte were essential for this transformation (Table 1, entries 16 and 17).

### Substrate scope

With optimal conditions in hand, we subsequently investigated the substrate scope and generality of this efficient electrochemical C−H deuteration transformation. As shown in Fig. 2A, a range of aryl-/alkyl-

pyridines with diverse substituents were evaluated. The steric effect of the substituent exerted little impact on this transformation, for example, 2-phenylpyridines bearing *o*-, *m*-, or *p*-methyl substituent furnished the desired products in good to excellent yields and D-inc (**2**−**4**), as well as that bearing two methyl (**5**). Other substituents, such as -OMe (**6**), -$^tBu$ (**7**), -Ph (**8**), -$O^iPr$ (**9**), and -OPh (**10**) were tolerated smoothly. The substrates that bearing fluorine (**11**), trifluoromethyl (**12**) and carboxy (**13**) groups were compatible and worked successfully, providing the desired products with high efficiency. Moreover, the deuterated products from phenylpyridine with heteroaryl-based (S, O, N) groups were also obtained in excellent yields and D-inc (**14**−**17**). Moreover, alternate and multiple substituted substrates were all tolerated and reacted smoothly in the system, providing satisfactory results (**18**−**22**, up to 99% yields, 86%−> 99% D-inc). In addition to aryl pyridines, we also investigated diverse alkyl pyridine derivatives. A wide range of substrates with various electron-donating and electron-withdrawing groups could all react well with $D_2O$ and produce the corresponding deuterium products in 75% to >99% D-inc (**23**−**28**).

Quinoline is a crucial class of heterocycles that are widely found in natural products, pharmaceuticals, dyes, and materials[72–75]. Therefore, we tried to probe the scope of quinolines with diverse functional groups to further exhibit the applicability of this protocol (Fig. 2B). We found that quinolines bearing electron-neutral (**29**), -donating (**30**−**39**), and -withdrawing (**40**) functional groups performed well in this transformation, affording the corresponding products in good yields (94%−99%) with excellent D-inc (84%− > 99%) and selectivity. In addition, 7,8-Benzoquinoline (**41**) and acridine (**42**, vaccines against infection and allergy) were also successfully worked in this protocol, resulting in satisfying results.

More importantly, we turned our attention to *N*-ligands (Fig. 3A), various of them were also accommodated, leading to the desired molecules in both good yields and D-inc (**43**−**50**), including bipyridine (**43**−**47**, **45**, Abametapir, a pediculicide for head lice infestation), benzimidazole (**48**) and phenanthroline (**49**−**50**), which indicated great compatibility and practicability of the protocol. It provided a new method for late-stage functional modification of *N*-ligands.

We next explored the late-stage deuteration of biorelevant compounds and some pharmaceutical molecules with this electro-reductive method (Fig. 3B). Gratifyingly, pyridine compounds derived from Citronellol (**51**), Phytol (**52**), Borneol (**53**), L-menthol (**54**), Carveol (**55**), D-galactopyranose (**56**), Picaridin (**57**), (S)-*N*-Boc-2-hydroxymethylmorpholine (**58**), Ibuprofen (**59**) and some pharmaceutical molecules, including Loratadine (**60**, Claritin, an anti-allergic drug), Abiraterone acetate (**61**, Zytiga, a prostate cancer drug) and Bisacodyl (**62**, Dulcolax, a laxative), all showed great compatibility and reactivity with this electro-reductive system, delivering the desired deuterated products in both excellent yields and D-inc%. These results indicated that this electrochemical protocol has great potential and prospect in application and modification of biorelevant compounds.

### Mechanistic studies

In order to further verify the rationality of this transformation, we performed a series of experiments to investigate the mechanism (Fig. 4). Firstly, we explored the effect of electrolytes (Fig. 4A). As expected, no product **1** was observed when NaCl, $LiBF_4$ or $KPF_6$ was employed as the electrolyte (Fig. 4A, entries 1−3). However, when we used other electrolytes containing $^nBu_4N^+$ ion, excellent D-inc of **1** was obtained (Fig. 4A, entries 4−9). Meanwhile, various anions from salts led to different yields (yield of **1** or recovered **S1**). These results illustrated that $^nBu_4N^+$ ion was crucial for this electrochemical reaction. Moreover, under the optimized conditions, *N*-butyl-2-phenylpyridinium iodide (**S1-a**, detected by HRMS, 212.1434), $^nBuI$ and $^nBu_3N$ were afforded in the absence of $D_2O$, which further demonstrated the important role of $^nBu_4N^+$ (Fig. 4B). Furthermore, we speculated that **S1-a** might be a key intermediate in this process. Hence, $^nBu_3N$ (1.0 equiv.)

**Table 1 | Screening of reaction conditions[a]**

S1 + $D_2O$ (50.0 equiv.) → [GF (+) | (−) Pb, $^nBu_4NI$ (1.0 equiv.), DMF, RT, 10 h, CCE = 20 mA] → 1 (2-phenylpyridine, deuterated at 4-position)

| Entry | Variation from standard conditions[a] | Yield of 1 or recover S1 (%)[b] | 1 (D%)[c] |
|---|---|---|---|
| **1** | **None** | **99** | **>99** |
| 2 | $^nBu_4NBF_4$ instead of $^nBu_4NI$ | 83 | 99 |
| 3 | $Et_4NI$ instead of $^nBu_4NI$ | 99 | 90 |
| 4 | $LiClO_4$ instead of $^nBu_4NI$ | 99 | – |
| 5 | NaI instead of $^nBu_4NI$ | 99 | – |
| 6 | NaOAc (1.0 equiv.) | 99 | 16 |
| 7 | DMA as solvent | 99 | 82 |
| 8 | MeCN as solvent | 99 | <5 |
| 9 | CF (+) | (−) Pb | 90 | 80 |
| 10 | GF (+) | (−) GF | 99 | 35 |
| 11 | GF (+) | (−) Pt | 99 | 53 |
| 12 | 10 mA | 99 | 60 |
| 13 | 15 mA | 99 | 81 |
| 14 | $T = 50\,°C$ | 99 | 99 |
| 15 | Ar | 99 | 99 |
| 16 | w/o electricity | 99 | -- |
| 17 | w/o electrolyte | 99 | -- |

Bold formatting shows that entry 1 is the optimal reaction conditions.

*CF* carbon felt, *DMF N,N*-dimethylformamide, *DMA N,N*-dimethylacetamide.

[a]Reaction conditions: undivided cell, graphite felt (GF) as anode, lead plate (Pb) as cathode, constant current at 20 mA, 2-phenylpyridine S1 (0.3 mmol), $D_2O$ (15.0 mmol, 50 equiv.), $^nBu_4NI$ (1.0 equiv.), DMF (4.0 mL), room temperature, air, 10 h.

[b]Isolated yield.

[c]Deuterium incorporation percentages were determined by $^1H$ NMR spectroscopy.

instead of $^nBu_4NI$, then a catalytic amount (5 mol%) of **S1-a~S1-d** were added to 2-phenylpyridine (**S1**) under standard conditions, furnishing the product **1** with >99% D (**S1-a**, $^nBu^+$), 96% D (**S1-b**, $Pr^+$), 90% D (**S1-c**, $Et^+$) and 87% D (**S1-d**, $Me^+$) respectively (Fig. 4C, a). On the other hand, the corresponding electrolytes were employed for the model reaction with standard conditions and gave compound **1** with >99% D ($^nBu_4NI$), 95% D ($Pr_4NI$), 90% D ($Et_4NI$), and 87% D ($Me_4NI$) severally (Fig. 4C, b). The results were almost identical to the previous ones. Next, we conducted several cyclic voltammetry (CV) studies (for details, please see the Supplementary Information on page 32). CV experiments on *N*-butyl-2-phenylpyridinium iodide (**S1-a**) gave a reductive peak at $E_{p/2} = −1.71\,V$ (−0.401 mA) vs. $Ag/Ag^+$ under Ar atmosphere (Fig. 4D, a, green line). An obvious reversible oxidative peak at $−2.65\,V$ (0.246 mA) vs. $Ag/Ag^+$ and a reversible reductive peak of **S1** at $−3.14\,V$ (−1.038 mA) vs. $Ag/Ag^+$ were observed under Ar atmosphere (Fig. 4D, a, red line). After mixing the two ingredients, the reductive peak of **S1-a** moved to $−1.70\,V$ (−0.453 mA) vs. $Ag/Ag^+$ (Fig. 4D, a, purple line). The oxidative and reductive peaks of **S1** changed to $−2.68\,V$ (0.142 mA) vs. $Ag/Ag^+$ and $−3.19\,V$ (−1.474 mA) vs. $Ag/Ag^+$ respectively (Fig. 4D, a, purple line). When $D_2O$ was added, the reductive peak of **S1-a** reduced to $−1.68\,V$ (−0.254 mA) vs. $Ag/Ag^+$ (Fig. 4D, b, blue line). The reductive peak of the mixture changed from $−3.19\,V$ (−1.474 mA) to $−3.27\,V$ (−1.766 mA) vs. $Ag/Ag^+$ under Ar atmosphere (Fig. 4D, b, blue line). However, the oxidative peak of the mixture was fully disappeared (Fig. 4D, b, blue line). All the results showed that a catalytic current was generated because of **S1-a**, which promoted the success of the protocol. Additionally, control experiments and CV studies also proved the bifunctional participation of $^nBu_4NI$ including the improvement of conductivity and the synthesis of **S1-a** in this protocol.

To show the robustness and utility of this reaction, we attempted to carry out a 10 mmol gram-scale experiment and a 100 mmol scale experiment to synthesize the product **1** (Fig. 4E). Surprisingly, the reaction maintained good selectivity with excellent yield and D-inc% (14.2 g, 91%, >99%D), which illustrated the high superiority and efficiency of this electroreductive protocol (For details, please see the Supplementary Information on pages 14−15). Meanwhile, we realized an elctrochemical continuous-flow reaction using a flow rate of 0.6 mL/min and a residence time for 3 h. The product **1** was obtained in excellent yield and D-inc% (Fig. 4F), which further demonstrated the potential application of this transformation. Then, we used several deuterated products to perform D/H exchange experiments with $H_2O$ (Fig. 4G), as shown, all D-molecules provided the corresponding initial materials in high yields (Fig. 4G, **S1, S6, S8, S16, S19** and **S39**, 95% H-> 99% H). It indicated that this electrochemical C−H deuteration transformation was reversible. Next, several competition experiments were performed (Fig. 4H). When this system was carried out in a mixture of $D_2O/H_2O$ (1/1, 7.5 mmol/7.5 mmol), only 16 % D of product **1** was produced (Fig. 4H, a), when the **S1** was replaced by **1** under the same conditions, 24 % D-inc% of **1** was afforded (Fig. 4H, b), which demonstrated that the ability of pyridine anions to capture $H^+$ is better than $D^+$. In addition, more experiments were conducted to explore the H/D exchange and D/H exchange rate (For details, please see the Supplementary Information on pages 18−20). In addition, a mixture of two substrates bearing diverse substituents performed different results under the same conditions (Fig. 4H, c, d). For example, it was obvious that the D-inc % of products that bearing -F was superior to that bearing -OMe.

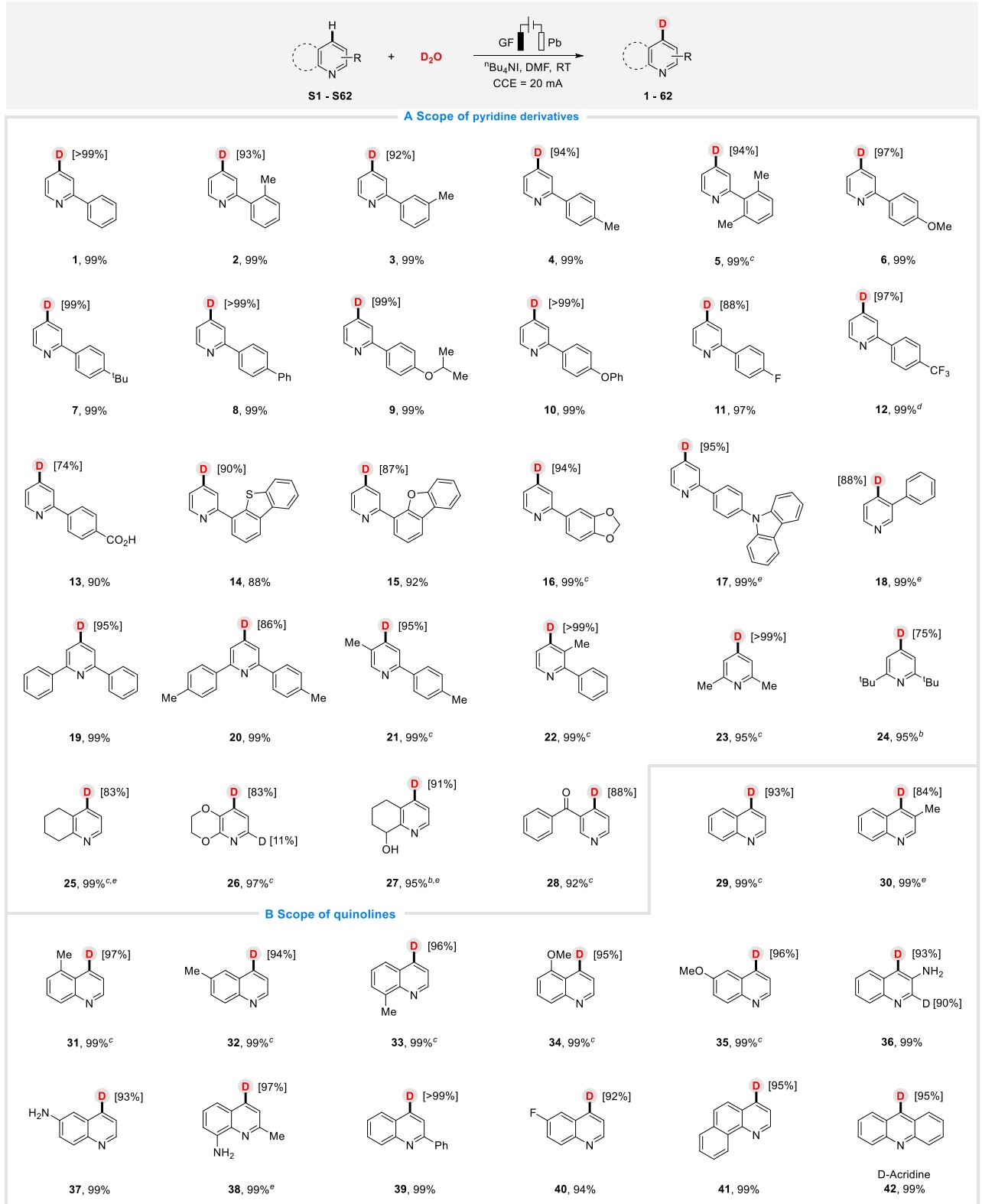

**Fig. 2 | Substrate scope. A** Pyridine derivatives. **B** Quinolones. Reaction conditions: [a]Electrochemical C−H deuteration of pyridines and quinolones in an undivided cell, GF as anode and Pb as cathode, constant current (20 mA), pyridine derivatives (0.3 mmol), D$_2$O (15.0 mmol), $^n$Bu$_4$NI (1.0 equiv.), DMF (4.0 mL), room temperature, 10 h, isolated yield. Deuterium incorporation percentages were determined by $^1$H NMR spectroscopy. [b]The reaction was conducted under 25 mA constant current. [c]30 mA. [d]40 mA. [e]16 h.

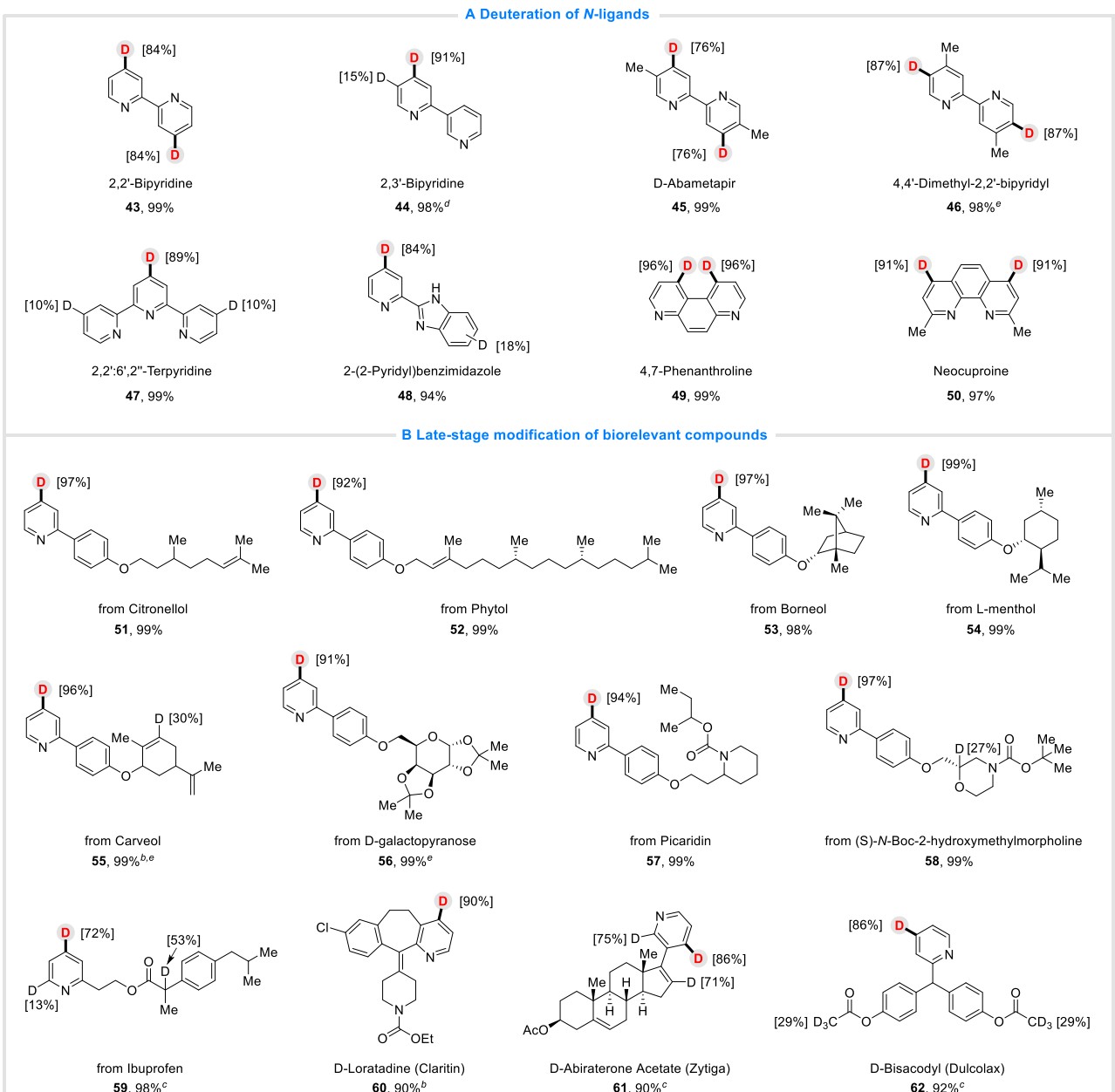

**Fig. 3 | Substrate scope. A** *N*-ligands. **B** Biorelevant compounds. Reaction conditions: [a]Electrochemical C−H deuteration of pyridines and quinolones in an undivided cell, GF as anode and Pb as cathode, constant current (20 mA), pyridine derivatives (0.3 mmol), D₂O (15.0 mmol), ⁿBu₄NI (1.0 equiv.), DMF (4.0 mL), room temperature, 10 h, isolated yield. Deuterium incorporation percentages were determined by ¹H NMR spectroscopy. [b]The reaction was conducted under 25 mA constant current. [c]30 mA. [d]40 mA. [e]16 h.

## Cyclic voltammetry studies

To gain further exploration into the reaction mechanism, in-depth studies were carried out through detailed cyclic voltammetry studies. An obvious reversible oxidation peak at −2.65 V (0.238 mA) vs. Ag/Ag⁺ and a reversible reduction peak of **S1** at −3.14 V (−1.025 mA) vs. Ag/Ag⁺ were observed under Ar atmosphere (Fig. 5, red line). In the presence of D₂O, the reversible oxidation peak at −2.65 V (0.238 mA) vs. Ag/Ag⁺ disappeared and the reduction peak shifted from −3.14 V (−1.025 mA) to −3.18 V (−1.204 mA) vs. Ag/Ag⁺ (Fig. 5, blue line). In the mixture of **S1** and ⁿBu₄NI, the reductive current of **S1** increased slightly, which might be attributed to the slight variation in conductivity (Fig. 5, green line). Then, D₂O was added to the mixed solution and the oxidation peak of **S1** disappeared again (Fig. 5, dark blue line), which was consistent with the previous results. However, since no desired product **1**

was detected with standard conditions in the absence of ⁿBu₄NI, it illustrated that ⁿBu₄NI played a crucial role in promoting the reactivity (Table 1, entry 17). Notably, we conducted CV experiments on **S1** with various scan rates (Fig. 5c), and the linear fit analysis also explained that a diffusion-control process might be involved in the conversion (Fig. 5d).

Based on the mechanistic experiments and CV studies, we proposed a possible mechanism (Fig. 6). Firstly, ⁿBu₄NI (**I**) splits into ⁿBu₃N (**II**) and ⁿBuI (**III**). Then ⁿBu₃N (**II**) was oxidized to radical cation **IV** on the anode to offer electrons. Next, ⁿBuI (**III**) forms a complex with **S1** to afford intermediate **V** (**S1-a**). Subsequently, **V** (**S1-a**) undergoes a single electron transfer on the cathode, to form the radical intermediate **VI**. Another single electron reduction generates the anion intermediate **VII**. The unique and high regioselectivity might be determined by these

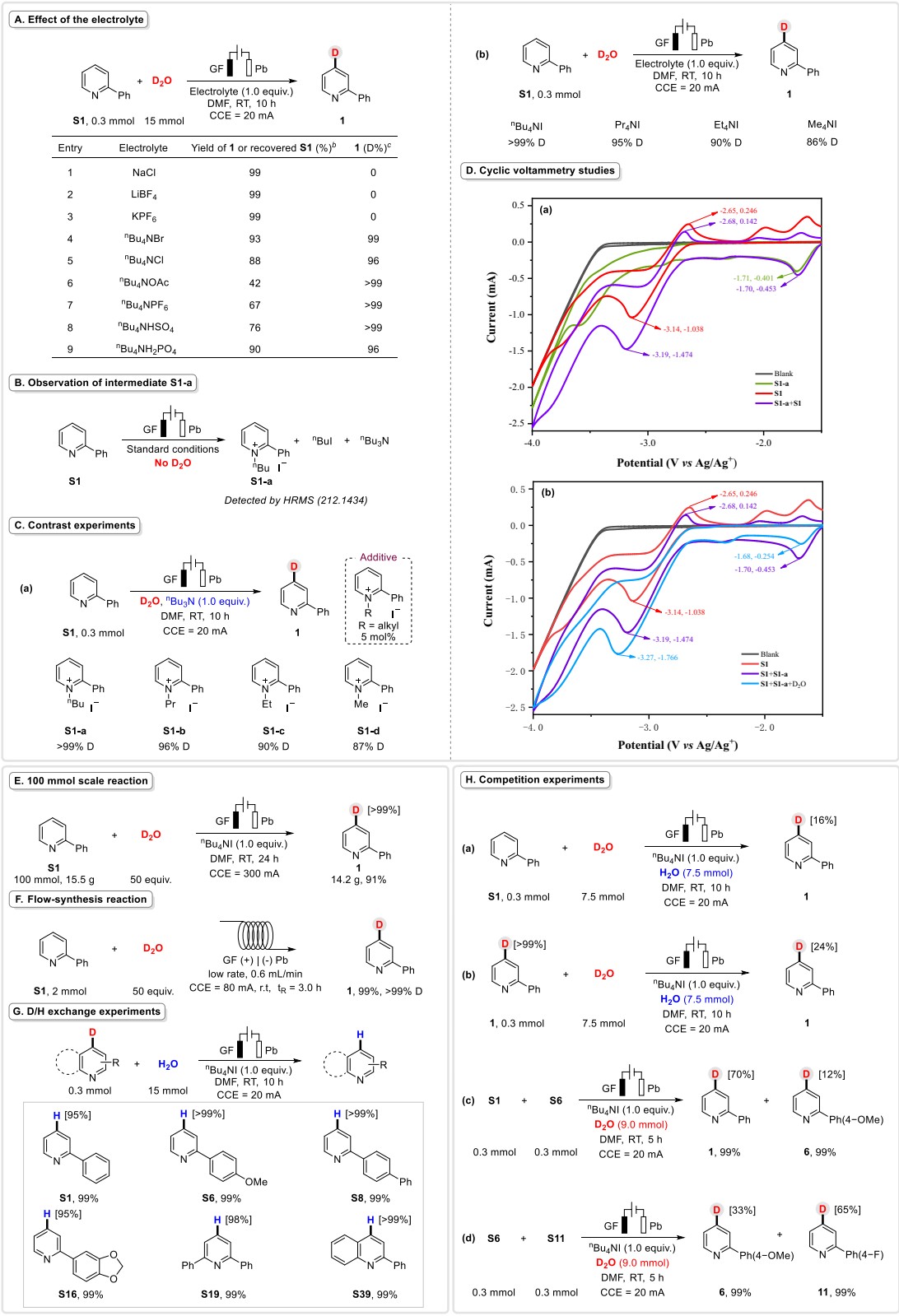

**Fig. 4 | Mechanistic studies. A** Effect of the electrolyte. **B** Observation of intermediate **S1-a**. **C** Contrast experiments. **D** Cyclic voltammetry studies, using glass carbon as work electrode, Pt plate and Ag/Ag⁺ as counter and reference electrodes. Scan rate: 100 mV s⁻¹. Solvent: DMF/ⁿBu₄NBF₄ (0.1 M) or MeCN/KPF₆ (0.1 M), **S1**

(0.01 M), D₂O (0.5 M), **S1-a** (0.001 M). Experiments were conducted under Ar unless otherwise noted. **a** CVs of **S1-a**, **S1** and the mixture of them. **b** CVs of **S1**, **S1** with **S1-a** and both of them with D₂O. **E** 100 mmol scale reaction. **F** Flow-synthesis reaction. **G** D/H exchange experiments. **H** Competition experiments.

**Fig. 5 | Cyclic voltammetry experiments.** Using glass carbon as work electrode, Pt plate and Ag/Ag+ as counter and reference electrodes. Scan rate: 100 mV s$^{-1}$. Solvent: DMF/$^n$Bu$_4$NBF$_4$ (0.1 M) or MeCN/KPF$_6$ (0.1 M), **S1** (0.01 M), D$_2$O (0.5 M), $^n$Bu$_4$NI (0.01 M). Experiments were conducted under Ar unless otherwise noted. **a** CVs of **S1** and **S1** with D$_2$O. **b** CVs of **S1, S1** with $^n$Bu$_4$NI and the reaction system. **c** CVs of **S1** performed at variable scan rates ranging from 10 mV s$^{-1}$ to 100 mV s$^{-1}$. **d** Linear fit analysis of $\upsilon_{scan}^{1/2}$ and $i_p$.

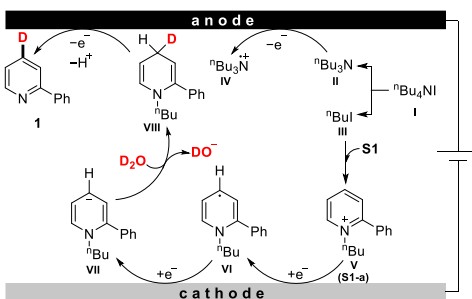

**Fig. 6 | Proposed reaction mechanism.** Anode: oxidative reaction. Cathode: reductive reaction. **S1** (2-phenylpyridine).

procedures. Subsequently, the anion intermediate **VII** reacts with D$_2$O and produces the deuteration intermediate **VIII**. Finally, intermediate **VIII** was oxidized on the anode, affording the target product **1**.

## Discussion

In conclusion, we reported the direct and efficient C4-selective deuteration of pyridine derivatives via the mode of electro-reductively driven C−H functionalization with D$_2$O at room temperature, without any metal, acid, and base. This transformation proceeded smoothly, as demonstrated with a wide range of substrates, forming the desired products with high regioselectivity and good to excellent D-incorporation. The utility of this protocol was also shown in the

synthesis of deuterated *N*-ligands and late-stage modification of bior-elevant compounds. Moreover, the mechanistic experiments and CV studies showed that *N*-butyl-2-phenylpyridinium iodide promotes the success of the electroreductive C−H deuteration procedure. Further electrochemical transformation and applications of pyridine salts and derivatives are ongoing in our laboratory.

## Methods

### General procedure of electroreductive C−H deuteration of pyridine derivatives

The electrocatalysis was carried out in an undivided cell with graphite felt (GF, 10 mm × 15 mm × 5 mm) as anode and Pb (10 mm × 15 mm × 0.3 mm) as cathode. To an oven-dried undivided electrochemical cell (15 mL) equipped with a magnetic bar was added organic *N*-heteroarenes (0.3 mmol, 1.0 equiv.), $^n$Bu$_4$NI (0.3 mmol, 110.8 mg, 1.0 equiv.) and D$_2$O (15 mmol, 300 mg, 50.0 equiv.), then anhydrous DMF (4.0 mL) was added via a syringe. The electrocatalysis system was performed at 20−40 mA of constant current for 10 h at room temperature. After that, the reaction mixture was extracted with EtOAc (30 mL × 3) and the combined organic phase was dried by anhydrous MgSO$_4$, filtered, and concentrated in vacuo. The crude product was purified by column chromatography to furnish the deuterated products.

## Data availability

The authors declare that the data supporting the findings of this study are available within the article and its Supplementary Information files.

Extra data are available from the author upon request. Source data are provided with this paper.

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

## Acknowledgements

Financial support from the National Key R&D Program of China (2023YFA1507203), National Natural Science Foundation of China (Grant No. 22371149, 22188101), the Fundamental Research Funds for the Central Universities (No. 63223015), Frontiers Science Center for New Organic Matter, Nankai University (Grant No. 63181206), and Nankai University are gratefully acknowledged.

## Author contributions

Y.Q. supervised the project, and provided guidance on the project. Y.Q. and Z.W.Z. conceived and designed the study and wrote the manuscript. Z.W.Z., R.Z., Y.L., Q.W. and Y.Q. performed the experiments, mechanistic studies, and revised the manuscript. Z.L.Z. conducted DFT calculations. All authors contributed to the analysis and interpretation of the data.

## Competing interests

The authors declare no competing interests.
