## [Peer Review File · Nature Communications]

Electrochemical C–H Deuteration of Pyridine Derivatives with D₂OReviewers' Comments:

Reviewer #1:

Remarks to the Author:

The authors present an innovative electrochemical approach for H/D exchange, utilizing D₂O as the deuterium source, to prepare deuterium-incorporated pyridines. This straightforward yet effective method has broad applicability for a variety of pyridine-based structures. Given the significance of pyridine and its related structures in both chemistry and biology, the creation of their deuterated counterparts is of considerable importance. In this context, this research is highly pertinent and suitable for publication in Nature Communications.

The observed regioselectivity is particularly intriguing as most reactions predominantly took place at the C4 position. However, when this position was obstructed, the reaction shifted to the C3 position (as seen in compound 46). As regioselectivity is a crucial aspect of this methodology, it would be beneficial for the authors to provide a more detailed explanation of these observations.

Reviewer #2:

Remarks to the Author:

The manuscript by Youai Qiu et al. describes a straightforward, metal-free, and acid-/base-free electrochemical C4-selective C–H deuteration of pyridine derivatives with economic and convenient D₂O at room temperature. This mild strategy has good chemo- and regioselectivity, exhibiting good-reaction scope for D-incorporation of pyridine-containing compounds. A reasonable mechanism has also been proposed on the basis of cyclic voltammetry studies. The manuscript might be suitable for publication on Nature Communications after carefully addressing the following concerns:

1. Is it possible to produce the deuterated pyridine products when utilizing 4-aryl/alkyl pyridines?
2. Different electrodes have a significant impact on the D-incorporation, and what could be the potential reasons for this phenomenon?
3. The reaction scope is highly restricted to selective deuteration of C4 position of pyridine, which limits its applicability in practical pharmaceutical applications. deuteration of other N-heterocyclic compounds should be evaluated, such as C4-substituted pyridines, pyrimidines, pyrroles, indoles, pyrazines etc. to greatly extended the reaction scopes and applicability. For example, if using C-4 substituted 59 compounds, will the D-incorporation reaction could be fully occurred on C-2 position?
4. The electrochemical C–H deuteration transformation was reversible via the same intermediate VIII. The H/D exchange and D/H exchange rate should be carefully studied to explain the reversible results like why the intermediate VIII converts to D-pyridines in D₂O, but convert to H-pyridines in H₂O.
5. Large scale and flow-synthesis should be demonstrated to improve its applicability.
6. More examples on Late-functionalization of commercial drugs should be carefully studied.

The point-by-point response to the reviewers' comments

Reviewer 1

Question: The observed regioselectivity is particularly intriguing as most reactions predominantly took place at the C4 position. However, when this position was obstructed, the reaction shifted to the C3 position (as seen in compound 46). As regioselectivity is a crucial aspect of this methodology, it would be beneficial for the authors to provide a more detailed explanation of these observations.

Response: Thank you for the professional comments and suggestions. As the D-O bond is highly polarized in D₂O, the deuteration may take place on the most negative site of the aromatic ring anion intermediate (**VII**, according to our proposed mechanism). We performed DFT calculations on the corresponding *N*-alkylated closed-shell anion intermediates (**VII**) of substrate **1**, **18**, **29**, **46**, and analyzed their natural population analysis (NPA) charge distribution on aromatic carbon atoms (Figure R1). According to the calculated charge distribution, the most negative site the C4 position if unsubstituted (**1-VII**, **18-VII**, **29-VII**), indicating a C4-deuteration regioselectivity, which is in accordance with experimental results. If the C4-position is substituted (**46-VII**), under DFT predictions, the electronegativity of C3-position is greater than the C2-position, rendering a C3-regioselectivity. Therefore, these DFT calculations on NPA charges reveal that the deuteration regioselectivity may mainly depend on the charge distribution on the aromatic ring (details of theoretical calculation and the supplemental references were provided in the Supplementary Information, page S-68 and S-156).

Figure R1. The NPA charges of aromatic C atoms in the *N*-alkylated closed-shell anion intermediates (**1-VII**, **18-VII**, **29-VII**, **46-VII**).

Meanwhile, in order to explore the regioselectivity of 4-substituted pyridines, we further tested some substrates under the standard conditions (please see Figure R2 below). The reaction preferentially occurred at C3 position when the C4 position was obstructed, which is also in accordance with our calculation results. On the other hand, all these results proved that our proposed mechanism is feasible.

Figure R2. Substrate scope 1. Reaction conditions: ^aElectrochemical C–H deuteration of pyridines and quinolones in an undivided cell, GF as anode and Pb as cathode, constant current (20 mA), pyridine derivatives (0.3 mmol), D₂O (15.0 mmol), ⁿBu₄NI (1.0 equiv.), DMF (4.0 mL), room temperature, 10 h, isolated yield. Deuterium incorporation percentages were determined by ¹H NMR spectroscopy.

Reviewer 2

Question 1: Is it possible to produce the deuterated pyridine products when utilizing 4-aryl/alkyl pyridines?

Response: Thank you for the professional comments. According to your comments, we further to explore the substrate scope and chemical reactivity, we tried to investigate some 4-aryl/alkyl pyridines (please see Figure R3 below, **63–S-f**). Among them, only **63** (11%D) and **64** (43%D) can provide deuterated products. However, no D-labeled products were detected when we used 4-methylpyridine (**S-a**), 4-propylpyridine (**S-b**), 4,4'-trimethylenedipyridine (**S-c**), 4-methylquinoline (**S-d**), 4-phenylquinoline (**S-e**) and 2,4-diphenylpyridine (**S-f**) under the standard conditions.

Figure R3. Substrate scope 2. Reaction conditions: ^aElectrochemical C–H deuteration of pyridines and quinolones in an undivided cell, GF as anode and Pb as cathode, constant current (20 mA), pyridine derivatives (0.3 mmol), D₂O (15.0 mmol), ⁿBu₄NI (1.0 equiv.), DMF (4.0 mL), room temperature, 10 h, isolated yield. Deuterium incorporation percentages were determined by ¹H NMR spectroscopy. ^bThe reaction was conducted under 25 mA constant current. ^c30 mA.

Question 2: Different electrodes have a significant impact on the D-incorporation, and what could be the potential reasons for this phenomenon?

Response: Thanks for your professional comments. In addition to our previous electrodes screening, (please see Figure R4, entries 1–4), we further tried Ni as cathodic material and Fe, Pt as anode materials. As a result, deuteration of **1** decreased

significantly when Ni was used as cathode (entry 5). When Fe was used as anode, only trace amount of product **1** was obtained (entry 6). To our delight, 90 D-inc% of **1** was afforded when Pt was used as anode (entry 7). So, the electrodes material seems to be essential to the transformation. These results have been put in the Supplementary Information, page S8.

Entry	Variation	Yield % ^b	D-inc % of 1 ^c
1	None	99	>99
2	CF (+) (-) Pb	90	80
3	GF (+) (-) GF	99	35
4	GF (+) (-) Pt	99	53
5	GF (+) (-) Ni	99	22
6	Fe (+) (-) Pb	trace	--
7	Pt (+) (-) Pb	99	90

Figure R4. ^aReaction conditions: undivided cell, graphite felt (GF) as anode, lead plate (Pb) as cathode, constant current = 20 mA, 2-phenylpyridine **S1** (0.3 mmol), D₂O (15.0 mmol, 50.0 equiv), ^tBu₄NI (1.0 equiv.), DMF (4.0 mL), room temperature, air, 10 h. ^bIsolated yield. ^cDeuterium incorporation percentages were determined by ¹H NMR spectroscopy. CF = carbon felt. DMF = *N,N*-dimethylformamide.

On the basis of our experimental study, combining with the literature reports (please see Figure below, *Angew. Chem. Int. Ed.* **59**, 18866–18884 (2020)), we think that the higher hydrogen evolution potential of Pb may be the reason for this result (Pb: -0.85, -0.91; Pt: -0.27, -0.09; Ni: -0.32).

Electrode Material	H ₂ Evolution η^a /V		O ₂ Evolution η^c /V		Conductivity ^d	Electrode Material	H ₂ Evolution η^a /V		O ₂ Evolution η^c /V		Conductivity ^d
	H ₂ O	MeOH	H ₂ O	H ₂ O	S/cm (10 ⁴)		H ₂ O	MeOH	H ₂ O	H ₂ O	S/cm (10 ⁴)
Ag	-0.59 ^[92]	-0.46 ^[93]	-0.21 ^[92]	0.61 ^[94]	68.17	Ni	-0.32 ^[95]		0.61 ^[94]		16.23
Al	-0.58 ^[93]				41.37	Pb	-0.85, ^[95] -0.91 ^[93]		0.80 ^[94]		5.21
Au	-0.12 ^[92]	-0.20 ^[92]	0.96 ^[94]		48.76	Pd	-0.01, ^[92] -0.09 ^[96]	-0.01 ^[92]	0.89 ^[94]		10.22
Be	-0.63 ^[95]				33.11	Pt	-0.27, ^[92] -0.09 ^[93]	-0.19 ^[92]	1.11 ^[94]		10.42
Bi	-0.33 ^[92]	-0.32 ^[92]			0.93	Plat. Pt	-0.01 ^[92]	-0.01 ^[92]	0.46 ^[94]		
Cd	-0.99 ^[93]		0.80 ^[94]		14.71	Rh	-0.08 ^[93]				23.3
Co	-0.3-0.4 ^[98]		0.39 ^[94]		17.86	Sn	-0.81 ^[97]				8.70
Cu	-0.46, ^[99] -0.57, ^[95] -0.60 ^[93]	-0.32 ^[99]	0.58 ^[94]		64.81	Ta	-0.20, ^[92] -0.41 ^[95]	-0.36 ^[92]			8.20
Fe	-0.40 ^[93]		0.41 ^[94]		11.67	Tl	-0.61, ^[92] -1.05 ^[95]	-0.44 ^[92]			6.67
Ga	-0.63 ^b ^[100]				7.35	W	-0.11, ^[92] -0.27 ^[93]	-0.32 ^[92]			20.75
Hg	-1.04 ^[93]				1.04	S Steel	-0.42 ^e ^[101]		0.28 ^e ^[101]		1.40 ^[102]
In	-0.80 ^[95]				12.50	Graphite	-0.47 ^[93]		0.50 ^[94]		0.0003, 0.4, ^[103] 2.6 ^[104]
Mo	-0.13, ^[92] -0.30 ^[95]	-0.28 ^[92]			20.62	BDD	-1.5 ^f ^[105]		1-2 ^g ^[105-107]		0.000001-0.002 ^[108]
Nb	-0.65 ^[95]				6.58	GC (RVC)	-1.13 ^h ^[109]				0.02-0.10 ^[110]

Question 3: The reaction scope is highly restricted to selective deuteration of C4 position of pyridine, which limits its applicability in practical pharmaceutical applications. deuteration of other *N*-heterocyclic compounds should be evaluated, such as C4-substituted pyridines, pyrimidines, pyrroles, indoles, pyrazines etc. to greatly extended the reaction scopes and applicability. For example, if using C-4 substituted 59 compounds, will the D-incorporation reaction could be fully occurred on C-2 position?

Response: Thank you for the professional comments.

(1) Based on your suggestions, we further explored the substrates of pyrimidines, pyrroles, indoles, pyrazines and pyrazoles (please see Figure R5 below, **65–S-o**). As a result, the deuterated products were obtained (**65–69**), albeit with low efficiency, and trace amount of D-inc% were detected (**70–71**). No desired products could be obtained when the starting material (**S-g–S-o**) were conducted under the standard conditions.

Figure R5. Substrate scope 3. Reaction conditions: ^aElectrochemical C–H deuteration of pyridines and quinolones in an undivided cell, GF as anode and Pb as cathode, constant current (20 mA), pyridine derivatives (0.3 mmol), D₂O (15.0 mmol), ⁿBu₄NI (1.0 equiv.), DMF (4.0 mL), room temperature, 10 h, isolated yield. Deuterium incorporation percentages were determined by ¹H NMR spectroscopy. ^bThe reaction was conducted under 25 mA constant current. ^c30 mA. ^d40 mA. ^e16 h.

(2) To further explore the deuterated position of C-4 substituted pyridines in this transformation, we tried the following starting materials (Figure R6 below). Most compounds occurred the D-labeling on C-5 and C-6 positions in moderate or low D-inc% (10% D–43% D, **63–64**, **72–75**). However, no D-labeled products could be observed when single or multiple substituted pyridines were used in this system (Figure R6 below, **S-a–S-f**, **S-p**).

Figure R6. Reaction conditions: ^aElectrochemical C–H deuteration of pyridines and quinolones in an undivided cell, GF as anode and Pb as cathode, constant current (20 mA), pyridine derivatives (0.3 mmol), D₂O (15.0 mmol), ⁿBu₄NI (1.0 equiv.), DMF (4.0 mL), room temperature, 10 h, isolated yield. Deuterium incorporation percentages were determined by ¹H NMR spectroscopy. ^c30 mA. ^d40 mA. ^e16 h.

In conclusion, according to our substrate scope, control experiments and the proposed mechanism, the D-labeled occurred preferentially on C-4 position of pyridine derivatives, such as the 2-phenylpyridine (**S1**), followed by C-5 position, which may mainly depend on the charge distribution on the aromatic ring, this was consistent with our DFT calculations. (Details of theoretical calculation were provided in the Supplementary Information, page S10).

Question 4: The electrochemical C–H deuteration transformation was reversible via the same intermediate VIII. The H/D exchange and D/H exchange rate should be carefully studied to explain the reversible results like why the intermediate VIII converts to D-pyridines in D₂O, but convert to H-pyridines in H₂O.

Response: Thanks for the professional comments and suggestions. We tried our efforts to probe the H/D exchange rate and D/H exchange rate experiments. As shown, with

the increase of time (from 0 h to 4.0 h), the D-inc% of **1** increased (Figure R7 below, entries 1–9). Meantime, we also tested the D-labeled product **1** (>99 D-inc%) under the same conditions except the H₂O instead of D₂O (Figure R8 below). Similarly, the H-inc% was increasing (the D-inc% was decreasing) with the extension of time (Figure R8 below, entries 1–9). For the chart of correlation trend, the slope of the H/D exchange rate ($k' = 14.93 \pm 0.72$, Figure R9, red line) is lower than the D/H's ($k' = 17.33 \pm 1.97$, Figure R9, black line), which demonstrated that the ability of pyridine anions to capture H⁺ is better than D⁺ under the same conditions.

(1) H/D exchange experiments

Entry	Time (h)	Yield % ^b	D-inc % of 1 ^c
1	0	0	0
2	0.5	99	12
3	1.0	99	21
4	1.5	99	30
5	2.0	99	37
6	2.5	99	44
7	3.0	99	50
8	3.5	99	56
9	4.0	99	61

Figure R7. ^aReaction conditions: undivided cell, graphite felt (GF) as anode, lead plate (Pb) as cathode, constant current = 20 mA, 2-phenylpyridine **S1** (0.3 mmol), D₂O (15.0 mmol, 50.0 equiv.), ⁿBu₄NI (1.0 equiv.), DMF (4.0 mL), room temperature, air. ^bIsolated yield. ^cDeuterium incorporation percentages were determined by ¹H NMR spectroscopy. CF = carbon felt. DMF = *N,N*-dimethylformamide.

(2) D/H exchange experiments

Entry	Time (h)	Yield % ^b	H-inc % of S1 (D-inc% of 1) ^c
1	0	0	0 (>99)
2	0.5	99	17 (83)
3	1.0	99	32 (68)
4	1.5	99	44 (56)
5	2.0	99	54 (46)
6	2.5	99	61(39)
7	3.0	99	65 (35)
8	3.5	99	68 (32)
9	4.0	99	71 (29)

Figure R8. ^aReaction conditions: undivided cell, graphite felt (GF) as anode, lead plate (Pb) as cathode, constant current = 20 mA, 2-phenylpyridine **1** (0.3 mmol), H₂O (15.0 mmol, 50.0 equiv), ⁿBu₄NI (1.0 equiv.), DMF (4.0 mL), room temperature, air. ^bIsolated yield. ^cDeuterium incorporation percentages were determined by ¹H NMR spectroscopy. CF = carbon felt. DMF = *N,N*-dimethylformamide.

Figure R9. The H/D exchange and D/H exchange rate studies

(3) Competition experiments

In addition, when this system was carried out in a mixture of D₂O/H₂O (1/1, 7.5 mmol/7.5 mmol), only 16 % D of product **1** was produced (please see the Figure below, **1**), but when we replaced the **S1** with **1** under the same conditions, low D-inc% of **1** was afforded as well (please see the Figure below, **2**), which further to demonstrate that the ability of pyridine anions to capture H⁺ is better than D⁺ under the same conditions (The NMR spectra have been put in the Supplementary Information page S22 and S24)

(4) Proposed reaction mechanism

Combined with our proposed reaction mechanism (please see the Figure R10 below), when the intermediate **VII** generates, it will react with D_2O preferably and produce the deuteration intermediate **VIII**, because the amount of D_2O is much more than H_2O in the system (The anhydrous DMF used in our experiments was purchased from Energy Chemical, 99.9%, Extra Dry, with molecular sieves, $Water \leq 30$ ppm). Then intermediate **VIII** was oxidized and dehydrogenated on the anode, affording the target product **1**. In this process, although intermediate **VIII** may also form the starting material **S1**, the D-labeled **1** will be finally obtained in good D-inc after multiple H/D exchange reaction due to the 50 equivalents of D_2O was added.

Figure R10. Proposed reaction mechanism

Question 5: Large scale and flow-synthesis could be demonstrated to improve its applicability.

Response: Thanks for the comments and suggestions.

(1) To further illustrate the potential of this transformation, we further conducted the large scale reaction, to our delight, the reaction of **S1** (100 mmol) with D_2O afforded the corresponding deuterated product **1** in 91% yields and 99 D-inc% (Figure below, this result has been added in the manuscript, page 11, Fig. 4, e).

The electrocatalysis was carried out in an undivided cell with graphite felt (GF, 30 mm × 45 mm × 5 mm) as anode and Pb (30 mm × 45 mm × 0.3 mm) as cathode. To an oven-dried undivided electrochemical cell (1000 mL) equipped with a magnetic bar was added 2-phenylpyridine (**S1**, 100 mmol, 15.5 g, 1.0 equiv.), nBu_4NI (100 mmol, 36.94 g, 1.0 equiv.) and D_2O (100 g, 50.0 equiv.). Then anhydrous DMF (400 mL) was added *via* a syringe. The electrocatalysis system was performed at 300.0 mA of constant current for 24 h at room temperature. After that, the reaction mixture was divided into many parts and extracted with EtOAc (60 ml × 40) and the combined organic phase were dried by anhydrous $MgSO_4$, filtered, and concentrated in vacuo. The crude product was purified by column chromatography to furnish the desired deuterated product **1** (14.2 g, 91%, >99% D).

Fig. R11. | (a) Reaction apparatus and tools. (b) General reaction apparatus. (c) Reaction potentiometer.

(2) Then we tried an electrochemical continuous-flow reaction to synthesize the product **1**, using a flow rate of 0.6 mL/min and a residence time for 3 h, the product **1** was obtained in 99% yield and >99 D-inc% (Figure below). This result have been added in the manuscript, page 11, Fig. 4, f.

General procedure for the flow-synthesis electrolysis

The substrate **S1** (2.0 mmol, 0.3104 g), nBu_4NI (2.0 mmol, 0.7388 g) and D_2O (50.0 equiv., 2 g) in dry DMF under air was pushed *via* a micro syringe pump to pass through the flow electrolytic cell with a flow rate of 0.6 mL min^{-1} and a 80 mA current to react for 3 h. After that, the reaction mixture was extracted with EtOAc ($100 \text{ ml} \times 3$) and the combined organic phase were dried by anhydrous $MgSO_4$, filtered, and concentrated in vacuo. The crude product was purified by column chromatography to furnish the desired deuterated product **1**.

Design of the flow-synthesis reaction microreactor

The flow electrolysis cell is assembled using two aluminum bodies (**a**, $75 \text{ mm} \times 75 \text{ mm} \times 15 \text{ mm}$) with a groove ($50 \text{ mm} \times 50 \text{ mm} \times 3.0 \text{ mm}$). The cathode (**b**, middle and right) consists a piece of Pb foil ($50 \text{ mm} \times 50 \text{ mm} \times 0.30 \text{ mm}$ thickness) fixed on a stainless steel base ($49 \text{ mm} \times 49 \text{ mm} \times 3.0 \text{ mm}$). The anode (**b**, left), which is made of carbon base ($49 \text{ mm} \times 49 \text{ mm} \times 5.0 \text{ mm}$), is insulated from the aluminum body by silicone film. The anode and cathode are held apart by a fluorinated ethylene propylene (FEP) foil (**e**) of 0.1 mm thickness. The whole device is held together by steel screws and wing nuts. The reaction mixture flows in and out through inlet and outlet (**a**, left, red circle). The reaction setup is shown in Figure R12.

Figure R12. (a). Aluminum bodies with a groove. (b) Electrode materials. (c) FEP spacer. (d) Anode and cathode. (e) Reaction setup

Question 6: More examples on Late-functionalization of commercial drugs should be carefully studied.

Response: Thank you for the comment and suggestion. As shown in the original manuscript, some pharmaceutical molecules, including Acridine (**42**, vaccines against infection and allergy), Abametapir (**45**, a pediculicide for head lice infestation), Loratadine (**60**, Claritin, an anti-allergic drug), Abiraterone acetate (**61**, Zytiga, a prostate cancer drug) and Bisacodyl (**62**, Dulcolax, a laxative) had good results. Then, we did our best efforts to explore some pyridine derivatives that derived from Picaridin (**57**), (S)-N-Boc-2-hydroxymethylmorpholine (**58**) and Ibuprofen (**59**) under the optimal conditions, to our delight, they all delivered the corresponding products both in excellent yields and D-inc% (please see Figure R13 below, and these results have been put in the manuscript, page 8, Fig. 3). In addition, we also tested other commercial drug compounds with diverse sensitive functional groups, for example, Amlexanox (**S-q**) Boc-3-(2-pyridyl)-L-alanine (**S-r**), Enoxacin (**S-s**), Pirenzepine (**S-t**) and Imatinib (**S-u**). Unfortunately, no D-labeled products were obtained, although many conditions

were tested, which may due to the drug molecules bearing many functional groups were sensitive under our standard electroreductive conditions.

Figure R13. Reaction conditions: ^aElectrochemical C–H deuteration of pyridines and quinolones in an undivided cell, GF as anode and Pb as cathode, constant current (20 mA), pyridine derivatives (0.3 mmol), D₂O (15.0 mmol), ⁿBu₄NI (1.0 equiv.), DMF (4.0 mL), room temperature, 10 h, isolated yield. Deuterium incorporation percentages were determined by ¹H NMR spectroscopy. ^bThe reaction was conducted under 25 mA constant current. ^c30 mA. ^d40 mA. ^e16 h.

We highly appreciate the reviewers' thorough reading and professional comments/suggestions about our manuscript. We hope that this manuscript will be suitable for publication in *Nat. Commun.* after the revision.

Best regards,

Youai Qiu

Reviewers' Comments:

Reviewer #1:

Remarks to the Author:

The authors have responded to the reviewer's comments appropriately.

Reviewer #2:

Remarks to the Author:

The authors make great efforts in improving the quality of the manuscript and the reviewers' concerns have been nicely addressed. Thus, publication of the revised version on Nature Communications could be now strongly recommended.

The point-by-point response to the reviewers' comments

Reviewer #1 (Remarks to the Author):

The authors have responded to the reviewer's comments appropriately.

Reviewer #2 (Remarks to the Author):

The authors make great efforts in improving the quality of the manuscript and the reviewers' concerns have been nicely addressed. Thus, publication of the revised version on Nature Communications could be now strongly recommended.

We highly appreciate the reviewer's positive comments on our manuscript. We are sure that the quality of this work has been greatly improved according to these nice comments and wise suggestions. Thanks very much.